# Quantitative Detection of Chromium Pollution in Biochar Based on Matrix Effect Classification Regression Model

**DOI:** 10.3390/molecules26072069

**Published:** 2021-04-03

**Authors:** Mei Guo, Rongguang Zhu, Lixin Zhang, Ruoyu Zhang, Guangqun Huang, Hongwei Duan

**Affiliations:** 1College of Mechanical and Electrical Engineering, Shihezi University, Shihezi 832003, China; guomei901116@163.com (M.G.); rgzh_jd@163.com (R.Z.); Zhlx2001730@126.com (L.Z.); 2College of Agricultural Engineering, Jiangsu University, Zhenjiang 212013, China; 3Key Laboratory of Northwest Agricultural Equipment, Ministry of Agriculture and Rural Affairs, Shihezi 832003, China; ry248@163.com; 4Laboratory of Biomass and Bioprocessing Engineering, College of Engineering, China Agricultural University, Beijing 100083, China; huangguangqun@126.com

**Keywords:** biochar, matrix effect, unsupervised/supervised classification, classification regression model, LIBS

## Abstract

Returning biochar to farmland has become one of the nationally promoted technologies for soil remediation and improvement in China. Rapid detection of heavy metals in biochar derived from varied materials can provide a guarantee for contaminated soil, avoiding secondary pollution. This work aims first to apply laser-induced breakdown spectroscopy (LIBS) for the quantitative detection of Cr in biochar. Learning from the principles of traditional matrix effect correction methods, calibration samples were divided into 1–3 classifications by an unsupervised hierarchical clustering method based on the main elemental LIBS data in biochar. The prediction samples were then divided into diverse classifications of calibration samples by a supervised K-nearest neighbor (KNN) algorithm. By comparing the effects of multiple partial least squares regression (PLSR) models, the results show that larger numbered classifications have a lower averaged relative standard deviations of cross-validation (ARSDCV) value, signifying a better calibration performance. Therefore, the 3 classification regression model was employed in this study, which had a better prediction performance with a lower averaged relative standard deviations of prediction (ARSDP) value of 8.13%, in comparison with our previous research and related literature results. The LIBS technology combined with matrix effect classification regression model can weaken the influence of the complex matrix effect of biochar and achieve accurate quantification of contaminated metal Cr in biochar.

## 1. Introduction

Returning biochar to farmland has become a research hotspot in China, since it can improve the quality of cultivated land [1]. Because of the enrichment effect of heavy metals during crop growth and crop-straw pyrolysis, the heavy metals in crop straw-based biochar may exceed the carrying capacity of farmland soil, resulting in secondary pollution. Therefore, the International Biochar Initiative (IBI) specifically highlights the importance of the analytical characteristics and producer’s certification of biochar as a soil remediation agent [2]. However, there are only a small number of reports about the production quality standards of biochar, and it is necessary to quantitatively and qualitatively analyze heavy metals in biochar with reference to soil pollution risk control standards.

Several rapid detection methods, such as biochemical sensors [3,4], test paper detection [5,6], the indicator biological method [7], enzyme-linked immunosorbent assay [8,9] and spectral analysis [10,11,12], have been widely used for metal detection in the fields of industrial analysis [13,14], biomedical engineering [15], food safety [16,17,18] and environmental ecological pollution assessment [19]. This can not only reduce the limit of detection (LOD), but can also improve the sensitivity and detection efficiency. In comparison with the other methods, spectral analysis has the advantages of multi-elemental high-throughput rapid detection when it is integrated with multiple chemo-metrics or even artificial intelligence algorithms. However, it is urgent to develop a simpler spectral analysis instrument with little sample pretreatment to avoid the complex sample pretreatment of traditional analysis instruments [20,21,22].

Laser-induced breakdown spectroscopy (LIBS) is a new elemental analytical technology that uses the atomic or ionic spectra emitted when laser ablates the sample surface in the focal plane. However, it still faces challenges in the accurate quantitative analysis of complex matrixes such as those in agricultural and food samples. The matrix effect mainly affects the plasma parameters (temperature, electron number density, etc.) and the ablation amount of the samples. The matrix characteristics of complex samples have a great influence on the total ablation amount of laser-excited samples. The larger the concentration difference between the matrix (main elements) and the analytical element, the more serious the interference effects will be. Generally, the standard addition method [23], the matrix matching method [24] and the pre-separation/enrichment method [25] are used for matrix effect correction. The standard addition method is mainly implemented to add pure substances with a certain quality as internal standards to the mixture of analyzed samples, and calculate the content of the tested components according to the mass ratio, the spectral peak areas ratio and relative correction factors. However, the matrix matching method has similar interference to the analysis elements because of the similar principal components, and the interference of matrix or principal components can be deducted by matching method. It can be seen that the main components in complex matrix have a great influence on the spectrum of LIBS of the analytical element, and the matrix effect of the analytical element has a similar regularity because of the similar concentration and existing form of the main components.

To remove the cumbersome process of traditional matrix matching or standard adding, a more efficient method of matrix effect classification is proposed. It uses an unsupervised hierarchical clustering method [26] to classify calibration samples based on the main elemental LIBS data, and develops multiple regression models for different matrix classifications. Then, the prediction samples are divided into diverse classifications of calibration samples by supervised KNN algorithm [27], and a quantitative prediction is made based on the best matrix classification regression model.

## 2. Results and Discussion

### 2.1. Matrix Elemental Analysis

Figure 1 shows that the carbonized straw fiber still maintains the graphite flake morphology, indicating the existence of a large amount of carbon. The statistical analysis of energy dispersive spectrometer (EDS) shows that the main elements in biochar are carbon (C), oxygen (O) and potassium (K), and their contents are about 59.36%, 17.49% and 9.99%, respectively. These results show high consistency with the relevant literature [28,29]. However, the Cr content is difficult to semi-quantitatively characterize and analyze, since its trace concentration exceeds the detection limit of EDS. Therefore, the matrix effect of the three elements is studied and applied for the classification of biochar samples.

### 2.2. Spectral Classification of Matrix Effect

The spectra of C, O, K and Cr in three biochar samples derived from rice husk (3#), rice straw (54#) and corn stalk (18#) were in the range of 192–846 nm, as presented in Figure 2. Two biochar samples (3#, 54#) should have a similar analytical spectral intensity of Cr on account of their similar concentration. In addition, the analytical spectral intensity of sample 18# should be higher than that of biochar samples (3#, 54#) because of its higher Cr content. Conversely, this is not the case. Furthermore, there was an obvious difference in the spectral intensities of main elements (C193.03 nm, C247.84 nm, O777.1 nm, O777.29 nm, O844.64 nm, K 766.29 nm and K 769.79 nm) in these three samples. The reason for this may be that the analytical spectra of Cr were interfered with by the matrix effect of the main elements of C, O and K in biochar.

However, the main elements suffered from few matrix effects due to their insignificant relative deviations of concentration; it was attempted to use their LIBS spectra to divide the biochar samples into diverse classifications. As for the K element, its analytical spectra of K 766.29 nm and K 769.79 nm were easily affected by the self-absorption effect [30]. Therefore, spectra of C193.03 nm, C247.84 nm, O777.1 nm, O777.29 nm and O844.64 nm were employed for classifying biochar samples, since these were the two elements with the highest concentration. Here, the calibration samples were divided into 1–3 classifications using hierarchical clustering method, and the supervised classification of prediction samples was carried out by KNN algorithm based on the classified calibration samples. The results of 2 and 3 classifications are shown in Figure 3.

As for the 2 classifications of the biochar matrix, it can be observed from Figure 3 that samples 1–25 and 26–46 of the calibration set in the 1 classification were divided unsupervised into the first (C1) and second (C2) classification, respectively, while samples 47–53 and 54–60 of the prediction set in the 1 classification were divided supervised into C1 and C2, respectively, as shown in Table 1. As for the 3 classifications of the biochar matrix, similar results were obtained from Figure 3b, whereby samples 1–25, 26–39 and 40–46 were divided unsupervised into the first (c1), second (c2) and third (c3) classification, while 47–53, 54–58 and 59–60 samples were divided supervised into c1, c2 and c3, respectively. Moreover, the two supervised models of KNN both have a robust performance due to their classification accuracies of cross-validation (CACV), which are higher than 90%. This indicates that the unsupervised 2 and 3 classification of calibration samples can be used for supervised classification of prediction samples Table 1.

### 2.3. Classification Regression Model

The peak broadening wavebands of Cr 357.83 nm, Cr 359.38 nm, Cr 360.5 nm, Cr 425.43 nm, Cr 427.48 nm, Cr 428.26 nm, Cr 428.95 nm, Cr 520.42 nm, Cr 520.56 nm and Cr 520.86 nm as shown in Figure 2 were employed to develop multivariate classification regression model of PLSR. The results of classification regression models in 1–3 classifications are shown in Table 2. Prior to developing calibration model, the LIBS data of Cr were preprocessed using the same algorithm in order to evaluate the model effect fairly.

The 1 classification regression model of PLSR was initially developed using the calibration set of 1–46 samples, resulting in the RSDCV value of 18.53%, while the 2 classification regression models were developed using two calibration sets of samples 1–25 and 26–46, yielding the RSDCV values of 12.70% and 23.51%. The results show that the 1 classification regression model effect is more robust than that of the second model in the 2 classification regression model, but is inferior to that of the first model. This indicates that the concentration and occurrence form of C and O in samples 1–25 are more similar, and the matrix effect is weak, which may not need to be subdivided again; meanwhile, some samples with a greater degree of matrix effect still exist in samples 26–46, which may need a second division. It was verified that samples 1–25 were divided into both the first classification in the 2- and 3 classifications, and the second classification samples in the 2 classifications were subdivided into the second and third classification samples in the 3 classifications, as shown in Table 1. The subdivided samples were used to develop PLSR models, yielding a lower RSDCV values of 11.02% and 13.24% than that of the second model in the 2 classifications. The results show that 3 classifications can further weaken the matrix effect of 2 classifications. The averaged relative standard deviations of cross-validation set (ARSDCV) of classification regression models in 1–3 classifications were 18.53%, 18.11% and 12.32%, respectively. This indicates that a classifications with a larger number have a better calibration performance, and the 3 classification regression model has the best modeling effect. This verifies that matrix effect of samples with similar concentrations and occurrence forms of main elements may be consistent, which leads to regular changes in the analytical spectra of Cr, and thus the matrix noise can be removed by normal preprocessing algorithms and the model performance can be improved. Thus, the 3 classification regression model of matrix effect classification partial least squares (MEC-PLS) was employed to quantitatively predict Cr content in biochar, resulting in the averaged relative standard deviations of prediction set (ARSDP) value of 8.13% as shown in Table 3. 

The performance in this work was superior to that of Fu et al. [31] and Duan et al. [32], and was comparable to that of Wang et al. [33]. This may be explained on the basis that, on the one hand, sensitive variables were all extracted from the full spectrum by the modified iterative predictor weighting (MIPW), full spectrum correction and modified iterative predictor weighting (FSC-MIPW) and Lasso algorithms in the related literature. However, the emission lines of multiple elements were all employed to develop the calibration models for predicting the Cr content, but the sensitive variables with large weight coefficients varied in these three reports. This means that the prediction ability of these models may be poor since their sensitive variables may not be suitable for soils in different habitats. One the other hand, the related literature should have a more robust performance than this work since a larger concentration of Cr signifies a stronger signal and a larger signal-to-noise ratio. However, the complex matrix effect present in these samples cannot be reduced by these sensitive variable extraction algorithms, resulting in the high degree of noise in the LIBS analytical spectra of Cr. Moreover, the ARSDP value of the developed 3 classification regression model decreased by 9.28% in comparison with our previous work [34] at similar concentrations. The reason for this may be that the unsupervised/supervised classification methods can successfully divide the tested samples into different classifications, which can obviously weaken the influence of matrix effect on the analytical spectra of Cr. In addition, the predicted values of Cr content in biochar were all below the risk value of 150 mg/kg according to “the soil environmental quality control standard for agricultural land soil pollution risk (Trial)” (GB15618–2018). The performance of the 3 classification regression model in terms of measured vs. predicted values of Cr in biochar are plotted in Figure 4.

## 3. Experimental Design and Methods

### 3.1. LIBS Device

The Benchtop LIBS (TSI, Minnesota, USA) apparatus is equipped with a Q-switched Nd:YAG laser emitting at 1064 nm and operating at a maximum frequency and energy of 2 Hz and 100 mJ, with a pulse width of 10 ns. The detector is a seven-channel spectrometer charge coupled device (CCD) array, with a wavelength range of 187.78–982.29 nm and a spectral resolution of λ/∆λ = 12,291.

To reduce the influence of laser pulse energy fluctuation on spectral intensity, the laser energy and spot size is set to 30 mJ and 200 μm, the number of single-point laser repeated ablations is 3. To avoid bremsstrahlung, the delay time of detector relative to laser pulse is set to 0.7 μs. After spectral collection of multiple spots on the sample surface, the averaged spectrum was taken as the final spectrum of each sample.

### 3.2. Sample Preparation

Sixty biochar samples derived from varied materials of rice husk, rice straw and corn straw were collected from Nanjing Zhironglian Technology Co., Ltd. (Nanjing, China). These samples were crushed using a pulverizer (WKF-130 type, Weifang, China) and screened with a 75 µm sieve, the resulting samples were placed in valve bags for use. Prior to spectral acquisition, each crushed sample was fixed on an aluminum substrate using the double-sided tape tableting method [35] to avoid the problems of poor molding effect and laser ablation splash caused by traditional tableting method.

Semi-quantitative analysis was primarily implemented to determine the main elements in biochar by using the scanning technology of EDS (SDD3310, IXRF Systems, City Texas, USA) attached to SEM (SU3500, Hitachi, City Tokyo, Japan) [36]. Each sample was determined three times. Furthermore, the Cr content in samples was determined by ICP-MS (PE NexION 300, Waltham, USA) [37], as shown in Table 4. It was observed that the Cr content in derivations of rice straw and rice husk was similar, but it was lower than that in corn straw-based biochar. This may be attributed to the fact that corn straw has a better enrichment power of Cr than that of rice straw and rice husk. According to the “soil environmental quality-agricultural land soil pollution risk control standard (trial)” (GB15618–2018) in China, the Cr content in biochar derived from the three materials is lower than the farmland soil pollution risk control value of 150 mg/kg.

### 3.3. Matrix Effect Classification Regression Modeling and Evaluation Criteria

Matrix effects can be divided into physical effects, chemical effects and absorption enhancement effects between elements [38,39]. Physical effects are mainly caused by particle size and inhomogeneity, which can be weaken by crushing and tableting. Chemical effects are mainly due to the crystal structure of the analytical element, which is weak for LIBS technology, since the high-energy laser could instantly transform the crystalline state into a high-temperature plasma state. However, the absorption enhancement effects between elements refer to the phenomenon whereby the analytical elemental spectrum depends not only on its own concentration, but also on the properties and concentration of main elements in samples. Fortunately, it can be understood from principles of matrix matching method and the standard addition method that the main elements in a complex matrix have a great influence on the analytical elemental LIBS spectra. Moreover, a more similar concentration and occurrence form of the main elements may signify a more similar matrix effect on the analytical element. Therefore, a new matrix effect classification regression model is proposed in this study. The main elements of collected biochar samples were firstly determined by X-ray energy dispersive spectrometer, the LIBS data of which were employed to classify the calibration samples using the unsupervised hierarchical clustering method [26]. Similarly, the prediction samples were divided into various classifications of calibration samples on the basis of a supervised KNN algorithm [27]. Multiple classification regression models of PLSR [40] were developed and compared, of which the model with the best calibration performance was employed for quantitative prediction.

The performance of PLSR models was evaluated by the root mean square errors of cross-validation (RMSECV) and prediction (RMSEP) sets, relative standard deviations of cross-validation (RSDCV) and prediction (RSDP) sets [40]. The formulas of RMSECV, RMSEP, RSDCV and RSDP are, respectively:(1)$$\mathrm{RMSECV}=\sqrt{\frac{\sum_{1}^{m}{\left(y_{i,actual}-y_{i,predicted}\right)}^{2}}{m-1}}$$
(2)$$\mathrm{RMSEP}=\sqrt{\frac{\sum_{1}^{n}{\left(y_{i,actual}-y_{i,predicted}\right)}^{2}}{n-1}}$$
(3)$$\mathrm{RSDCV}\left(\%\right)=\frac{RMSECV}{\bar{y}}=\frac{\sqrt{\frac{\sum_{1}^{n}{\left(y_{i,actual}-y_{i,predicted}\right)}^{2}}{n-1}}}{\bar{y}}\times 100\%$$
(4)$$\mathrm{RSDP}\left(\%\right)=\frac{RMSEP}{\bar{y}}=\frac{\sqrt{\frac{\sum_{1}^{n}{\left(y_{i,actual}-y_{i,predicted}\right)}^{2}}{n-1}}}{\bar{y}}\times 100\%$$
where *m* and *n* are the sample numbers of calibration and prediction sets, respectively; and are the measured and predicted values, respectively; and is the averaged value of prediction set. A lower value of RSDCV (or a lower value of RMSECV) signifies a better modeling effect, while a lower value of RSDP (or a lower value of RMSEP) signifies a better prediction accuracy [40].

## 4. Conclusions

In this study, we investigated the feasibility of applying LIBS technology for the quantitative analysis of heavy metal Cr in biochar. To reduce the influence of complex matrix effects, calibration samples of biochar were divided unsupervised into 1–3 classifications using the main elemental LIBS data by hierarchical classification method. The prediction samples were divided supervised into diverse classifications of calibration samples by the KNN algorithm. In comparison with the other two classification regression models, the 3 classification regression model showed a more robust performance in modeling, which was finally employed for quantitative analysis of prediction set. Furthermore, the developed MEC-PLS models had a better prediction performance, with a lower ARSDP value of 8.13%, in comparison with our previous research and related literature results. The predicted values of Cr content in biochar were all below the risk value of 150 mg/kg (GB15618-2018). The results imply that the element of Cr in the produced biochar are within the carrying capacity of farmland soil, which will not trigger a secondary pollution. The results show that LIBS technology combined with matrix effect classification regression model was able to realize accurate and quantitative detection of Cr in biochar, providing a technical reference for the development of related portable or online LIBS detection equipment.

## Figures and Tables

**Figure 1 molecules-26-02069-f001:**
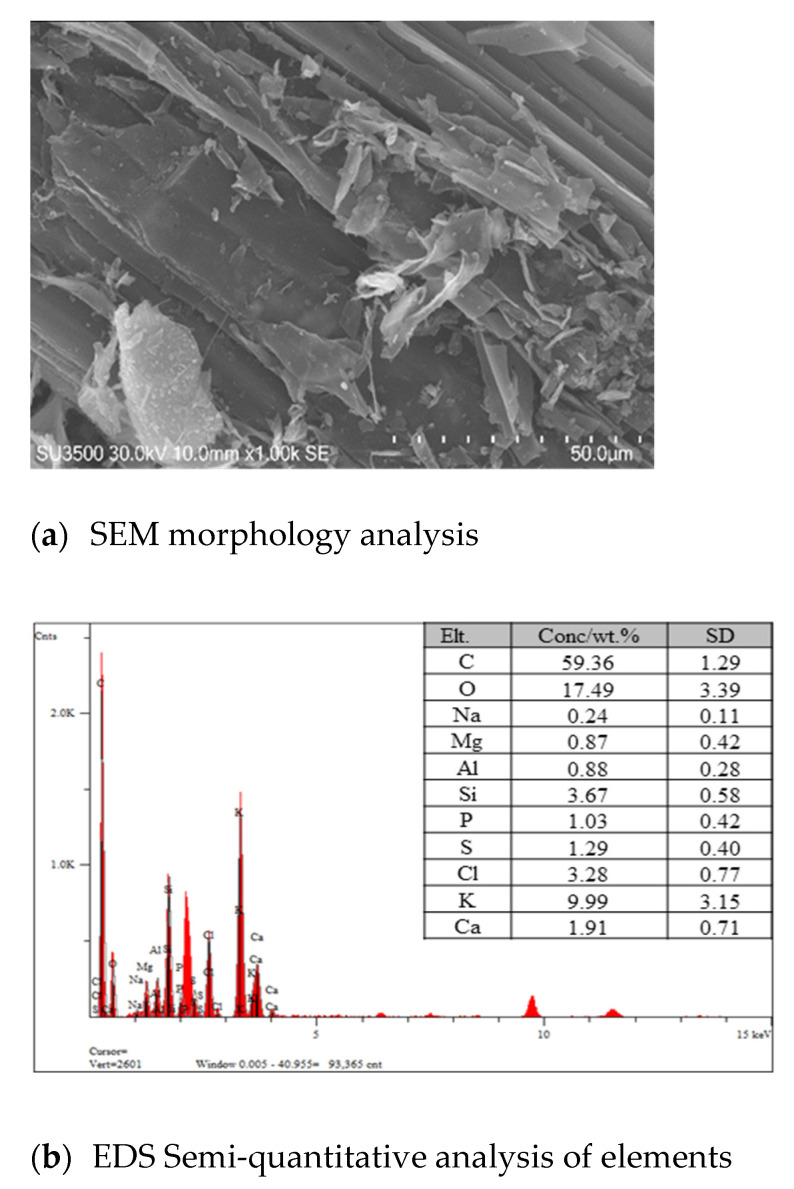
SEM (**a**) and EDS (**b**) analysis of biochar.

**Figure 2 molecules-26-02069-f002:**
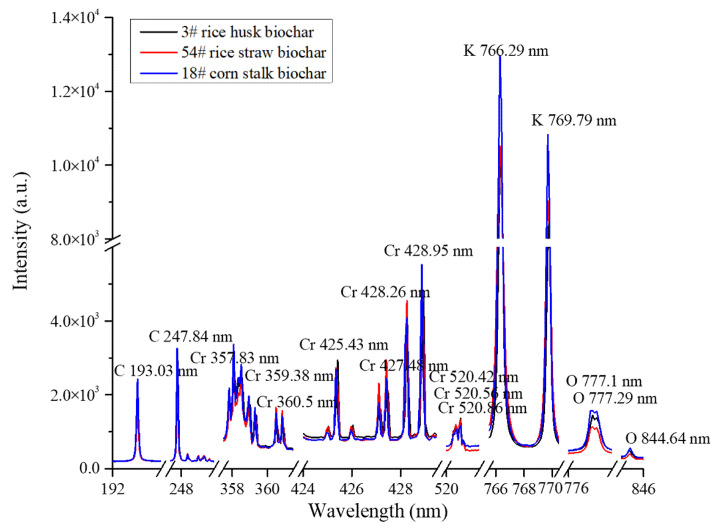
LIBS spectra of biochar.

**Figure 3 molecules-26-02069-f003:**
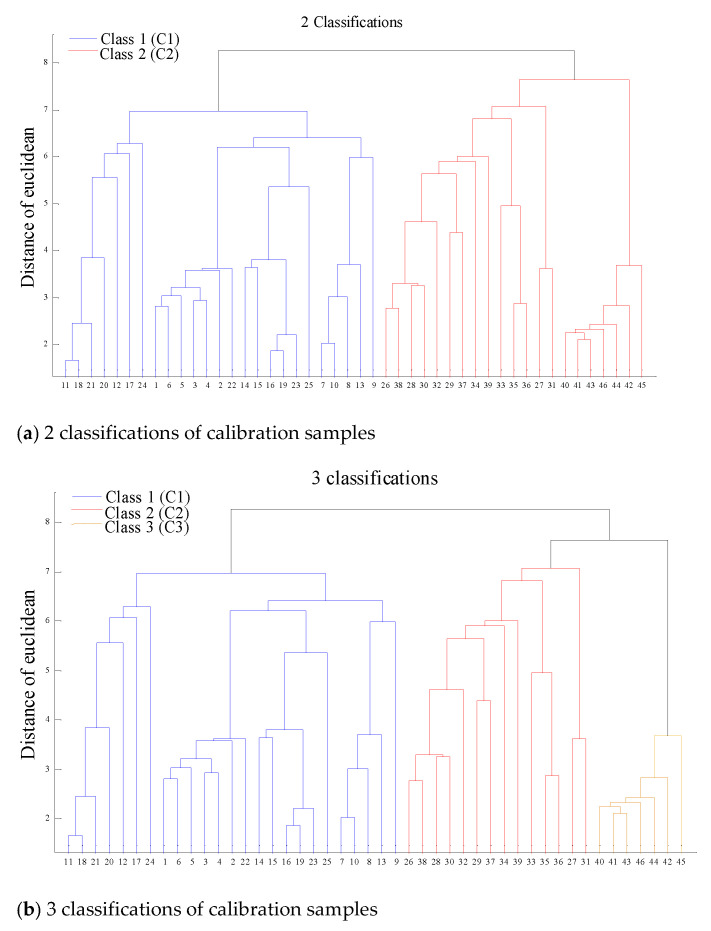
Unsupervised 2 classifications (**a**) and 3 classifications (**b**) hierarchy diagram of calibration samples.

**Figure 4 molecules-26-02069-f004:**
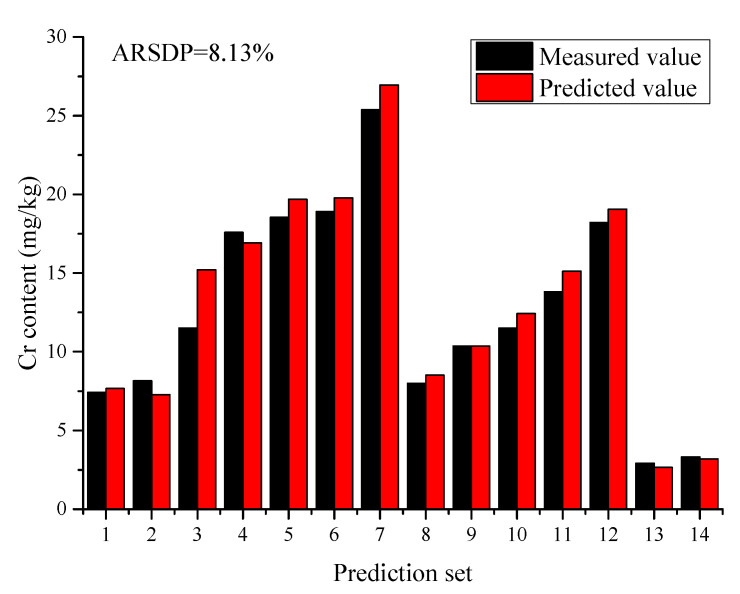
Prediction results of the 3 classification regression model.

**Table 1 molecules-26-02069-t001:** Classification results of biochar samples.

Main Matrix	2 classifications	3 Classifications
C+O (76.85%)	C1: 1–25 (calibration), 47–53 (prediction) C2: 26–46 (calibration), 54–60(prediction)	c1: 1–25 (calibration), 47–53 (prediction) c2: 26–39 (calibration), 54–58 (prediction) c3: 40–46 (calibration), 59–60 (prediction)

**Table 2 molecules-26-02069-t002:** Results of classification regression model.

Classifications	Classification	RMSECV (mg/kg)	RSDCV (%)	ARSDCV (%)
1	None	2.38	18.53%	18.53%
2	C1	1.99	12.70%	18.11%
	C2	2.23	23.51%	
3	c1	1.99	12.70%	12.32%
	c2	1.38	11.02%	
	c3	0.45	13.24%	

**Table 3 molecules-26-02069-t003:** Cr detection in biochar and soil in related literature.

Particle	Element	Range (mg/kg)	RSDP/ARSDP (%)	Remarks	Ref.
Biochar	Cr	2.92–25.38	8.13%	MEC-PLS ^1^	In this work
Soil	Cr	48–410	23.019	MIPW-PLS ^2^	Fu et al. 2017 [31]
Soil	Cr	48–410	17.673	FSC-MIPW-PLS ^3^	Duan et al. 2018 [32]
Soil	Cr	18.29–164.06	11.460	Lasso ^4^	Wang et al. 2018 [33]
Biochar	Cr	5.05–19.15	17.41%	PLS	Duan et al. 2019 [34]

^1^ MEC-PLS: matrix effect classification partial least squares; ^2^ MIPW-PLS: modified iterative predictor weighting–partial least squares; ^3^ FSC-MIPW-PLS: full spectrum correction and modified iterative predictor weighting–partial least squares. ^4^ Lasso: least absolute shrinkage and selection operator.

**Table 4 molecules-26-02069-t004:** Statistical results of Cr (mg/kg) content.

Derivation (#1~#15)	Content	Derivation (#16~#30)	Content	Derivation (#31~#45)	Content	Derivation (#46~#60)	Content
rice husk	7.37	corn stalk	18.21	rice straw	10.67	rice husk	4.5
rice husk	7.47	corn stalk	18.63	rice straw	11.47	rice husk	7.42
rice husk	7.83	corn stalk	18.88	rice husk	11.73	rice husk	8.15
rice husk	8.11	corn stalk	18.9	rice straw	12.81	rice husk	11.51
rice husk	8.22	corn stalk	18.98	rice husk	13.75	corn stalk	17.59
rice husk	8.53	corn stalk	19.15	rice husk	15.15	corn stalk	18.55
rice husk	10.59	rice husk	19.56	rice straw	16.47	corn stalk	18.91
rice husk	11.54	corn stalk	26.15	corn stalk	17.08	corn stalk	25.39
rice husk	12.11	corn stalk	26.34	corn stalk	20.15	rice straw	7.99
rice husk	12.52	corn stalk	28.51	rice straw	2.79	rice straw	10.36
corn stalk	15.27	rice straw	7.9	rice straw	2.9	rice straw	11.5
corn stalk	15.61	rice straw	8.37	rice straw	3.04	rice straw	13.81
rice husk	17.28	rice straw	9.32	rice straw	3.28	corn stalk	18.22
corn stalk	17.94	rice straw	9.9	rice straw	3.3	rice straw	2.92
corn stalk	18.03	rice straw	10.6	rice straw	3.98	rice straw	3.32

## Data Availability

Not applicable.

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
