# Peer review of "Quantitative Detection of Chromium Pollution in Biochar Based on Matrix Effect Classification Regression Model"

_molecules, 2021, doi:10.3390/molecules26072069_

Round 1

Reviewer 1 Report

Some parts in the methodology should be cited, because it is the same was used in previous authors work. Also, authors do not created a new methodology, it’s well known? Also:

107 line: What kind of information provid fig. 1? It should be revomed.

156-160 lines information should be removed to the methodology part.

The results and discussion section does not provide any discussion on the obtained results in terms of how they relate or differ from existing literature, or in what way they are novel (the comparison data with previuos data should be provided).

Also, 167 line: SEM data should be described and cited or removed. What about Ch content in the matrix element analysis?

180 line: what about determination wih LIBS spectra of other elements  which are presented in fig.2? Does LIBIS spectra present quantitative analysis data?

What about Quantitative Detection of Chromium Pollution in Biochar? Authors should present quantitative analysis data. For this reason, the scientific significance of article is questionable.

The work should bring important information about the  coexistence and mechanism of phase transformation.

By the way, conclusions of manuscript should be improved.

Author Response

Responds to the reviewer’s comments:

  1. Some parts in the methodology should be cited, because it is the same was used in previous authors work. Also, authors do not created a new methodology, it’s well known? Also:107 line: What kind of information provid Fig. 1? It should be revomed. 156-160 lines information should be removed to the methodology part.

Answer: Thank you for your valuable suggestions. Some parts in the methodology have been cited in the uploaded revised manuscript.

107 lines: Fig. 1 has been removed.

156-160 lines information has been removed to the methodology part.

  1. The results and discussion section does not provide any discussion on the obtained results in terms of how they relate or differ from existing literature, or in what way they are novel (the comparison data with previuos data should be provided). Also, 167 line: SEM data should be described and cited or removed. What about Cr content in the matrix element analysis? 180 line: what about determination with LIBS spectra of other elements which are presented in fig.2? Does LIBS spectra present quantitative analysis data?

Answer: The discussion on the obtained results (in terms of how they relate or differ from existing literature, or in what way they are novel) have been added in the uploaded revised manuscript.

Line 167: SEM data has been described and cited in the uploaded revised manuscript. However, the Cr content can hardly been semi-quantitatively characterized and analyzed in Fig. 2 since its trace concentration exceeds the detection limit of SEM-EDS.

Line 180: Many works [1-2] indicate that the physicochemical properties of plasma have a great influence on the emitted LIBS spectra, while the physicochemical properties of plasma are mainly determined by the collision and ionization of atoms and molecules in samples. Therefore, a larger amount of the elemental content signifies a greater influence of matrix effect on the LIBS spectra of the tested samples. Here, Carbon and oxygen account for almost 80% of the mass in biochar, indicating a greater influence of matrix effect on the LIBS spectra than the other elements in the tested samples. Therefore, it is better to use LIBS spectra of C and O to determine the classification of calibration and prediction samples. Furthermore, the peak broadening wavebands of Cr was used to develop the matrix effect classification regression models for the quantitative analysis.

[1] Hou, Z., Afgan, M. S., Sheta, S., Liu, J.; Wang, Z. Plasma modulation using beam shaping to improve signal quality for laser induced breakdown spectroscopy. Journal of Analytical Atomic Spectrometry, 2020, 35, 1671-1677.

[2] Sheta, S., Afgan, M. S., Jiacen, L., Gu, W., Hou, Z.; Wang, Z. Insights into Enhanced Repeatability of Femtosecond Laser-Induced Plasmas. ACS omega, 2020, 5, 30425-30435.

  1. What about Quantitative Detection of Chromium Pollution in Biochar? Authors should present quantitative analysis data. For this reason, the scientific significance of article is questionable.

Answer: Thank you for your valuable suggestions. We are sorry that we were unable to upload the test data before due to our negligence. Our another study is also using the matrix effect classification regression method to detect a variety of metal elements in biomass straw. The paper is being written. We look forward to your attention.

  1. The work should bring important information about the coexistence and mechanism of phase transformation. By the way, conclusions of manuscript should be improved.

Answer: It is hard to collect more important information about the coexistence and mechanism of phase transformation because we don't have more advanced and expensive equipments (ultra high speed camera, fs-LIBS, et.al.,) yet. We will take a further study to collect more information through the cooperation with relevant research institutes based on this work.

Conclusions of manuscript have been improved.

Reviewer 2 Report

The main objective of this work is the rapid detection of heavy metals in biochar from different materials, which can provide a guarantee that the contaminated soil will no longer withstand secondary pollution. Sixty samples of biochar derived from various materials of rice husks, rice straw and corn straw were tested. The EDX technique showed that the main elements in biochar are carbon (C), oxygen (O) and potassium (K), and their content is about 59.36%, 17.49% and 9.99%, respectively. Therefore, the matrix effect of the three elements is studied and applied for the classification of biochar samples. The Cr content was also determined by the LIBS technique and it was observed that the Cr content in rice straw and rice husk was similar, but was lower than that in maize straw biochar. The authors showed that the regression model with 3 classifications led to the best performances. I suggest the authors to check the proposed regression model for other experimental data that were not used in the modeling, in order to validate the obtained results.

Author Response

Responds to the reviewer’s comments:

  1. The main objective of this work is the rapid detection of heavy metals in biochar from different materials, which can provide a guarantee that the contaminated soil will no longer withstand secondary pollution. Sixty samples of biochar derived from various materials of rice husks, rice straw and corn straw were tested. The EDX technique showed that the main elements in biochar are carbon (C), oxygen (O) and potassium (K), and their content is about 59.36%, 17.49% and 9.99%, respectively. Therefore, the matrix effect of the three elements is studied and applied for the classification of biochar samples. The Cr content was also determined by the LIBS technique and it was observed that the Cr content in rice straw and rice husk was similar, but was lower than that in maize straw biochar. The authors showed that the regression model with 3 classifications led to the best performances. I suggest the authors to check the proposed regression model for other experimental data that were not used in the modeling, in order to validate the obtained results.

Answer: Thank you for your valuable suggestions. The proposed regression model for other experimental data that were not used in the modeling was checked and validated as presented in the following Table 1. The results of cross-validation set and prediction set both indicate that a larger number of classification has a better calibration performance, and the 3-classification regression model has the best modeling effect.

Table 1. Results of Classification Regression Model.

Classifications

Classification

RMSECV (mg/kg)

RSDCV (%)

ARSDCV (%)

RMSEP (mg/kg)

RSDP (%)

ARSDP (%)

1

None

2.38

18.53%

18.53%

1.59

12.67%

12.67%

2

C1

1.99

12.70%

18.11%

1.67

10.87%

11.60%

C2

2.23

23.51%

1.20

12.33%

3

c1

1.99

12.70%

12.32%

1.67

10.87%

8.13%

c2

1.38

11.02%

0.84

6.79%

c3

0.45

13.24%

0.21

6.74%

Round 2

Reviewer 1 Report

Accept after minor revision (corrections to minor methodological errors and text editing)

Reviewer 2 Report

Although the authors did not respond to all previous observations, they agree with the publication of this version of the manuscript.